# Preparation of Visible-Light Active Oxygen-Rich TiO$_2$ Coatings Using Low Pressure Cold Spraying

Anna Gibas [1,*], Agnieszka Baszczuk [1], Marek Jasiorski [1], Marcin Winnicki [2] and Daniel Ociński [3]

1   Department of Mechanics, Materials and Biomedical Engineering, Wroclaw University of Science and Technology, 25 M. Smoluchowskiego Street, 50-370 Wroclaw, Poland; agnieszka.baszczuk@pwr.edu.pl (A.B.); marek.jasiorski@pwr.edu.pl (M.J.)
2   Department of Materials Science, Strength and Welding Technology, Wroclaw University of Science and Technology, 5 Łukasiewicza Street, 50-371 Wroclaw, Poland; marcin.winnicki@pwr.edu.pl
3   Department of Chemical Technology, Wroclaw University of Economics and Business, 118/120 Komandorska Street, 53-345 Wroclaw, Poland; daniel.ocinski@ue.wroc.pl
*   Correspondence: anna.gibas@pwr.edu.pl

**Abstract:** Visible-light active photocatalysts in the form of coatings that can be produced using large-scale methods have attracted considerable attention. Here we show a facile approach to deposit coatings using the low pressure cold spray (LPCS) from oxygen-rich amorphous titanium dioxide, which is a structurally-unconventional feedstock powder for LPCS. We synthesized amorphous TiO$_2$, in which we introduced numerous defects, such as oxide groups (peroxy and superoxy) in volume and hydroxyl groups on the surface. Then we deposited as-prepared powder preserving the presence of active groups, which we demonstrated using Raman spectroscopy. To show the activity of the prepared coatings, we perform methylene blue degradation under visible light. Our research shows that it is worth considering the internal atomic structure and surface chemistry of the powders to be preserved after low pressure cold spraying.

**Keywords:** low pressure cold spraying; TiO$_2$; yellow TiO$_2$; visible-light-active photocatalyst; hydrogen peroxide

## 1. Introduction

The formation of TiO$_2$ photocatalytic coatings has been a constant challenge over the past 30 years [1–3]. During this time, titanium dioxide has remained one of the most photocatalytically efficient and highly stable materials, and stayed at a low price. TiO$_2$ coatings for photocatalytic systems may be produced through several low-temperature preparation techniques: by the transport of the precursor to the substrate using gas or plasma (chemical vapor deposition (CVD) [4], pulsed laser deposition (PLD) [5], atomic layer deposition (ALD) [6], ion sputtering [7]); deposition of liquid phase containing precursors (dip-coating [8], spin coating [9], spray pyrolysis [10]); or deposition of precursors in the solid-state (using screen-printing, doctor blade method [11] or thermal spraying [12]). Due to increasing environmental pollution, greener and more energy-efficient deposition methods that are large-scale and inexpensive are being actively sought. This urge does not change the main requirement of photocatalytic coatings, which is, namely, to retain or create a specific surface area as high as possible [13].

In response to those demands (apart from environmental safety and economic efficiency) the cold-spray method offers preservation of the nanoparticulate structure of the initial feedstock [14–16]. Several publications have already been published on various photocatalytic cold-sprayed coatings [12,15–17]. The coatings were cold sprayed most often using the high pressure variant (HPCS), >2 MPa [12], which have an environmental cost that consists of the consumption of gas and electrical power as well as the replacing of parts due to the relatively high rate of nozzle erosive wear [18]. Deposition at lower pressures

(below 1 MPa), in which the energy expenditure is reduced, if successful, proceeded so far with the use of helium or nitrogen as the carrier gas, just like in the HPCS variant [19]. Most cold-sprayed $TiO_2$ coatings concern commercially available crystalline $TiO_2$ feedstocks [19,20]. The most typical among them is Degussa (Evonik) P25, which is a mixture of rutile and anatase in a proportion of one to four [1,2]. Until now, we did not manage to find any successful approach to spray pure P25 without additions using the cold spray. Regardless of the deposition pressure, the deposition mechanism, and simultaneously, the main depositional problem of ceramic feedstock powder arises from its brittleness [21]. The impact of ceramic aggregates accelerated to sonic velocities causes the aggregate to break down, which then enables the interlocking of secondary particles. The as-formed layer is usually very thin as the incoming aggregates detach the already embedded particles [22]. In addition, too much energy (i.e., increased by the use of high pressure carrier gas) can lead to uncontrolled phase transformation (for instance of anatase to rutile, which is a less photocatalytically active polymorph) or grain overgrowth (decreasing particle surface area); while an insufficient amount energy impedes the embodiment of the particles [22]. The deposition efficiency in a cold spray can be enhanced by increasing the ductility of feedstock powder particles [16,17]. One method is to admix ductile phase to ceramic feedstock [17]. Another method is to spray amorphous $TiO_2$ [16], and this is the amorphous $TiO_2$ which seems, up to now, to be the only efficient variant for feedstock powder to deposit thick single-phase coatings using low pressure cold spray (LPCS). We showed in our previous work [16,23] that the synthesis of amorphous $TiO_2$ may, on the one hand, impart plasticity reducing erosion during the deposition process, and on the other hand favor the beneficial transformation in terms of photocatalytic efficiency (from the amorphous phase to anatase). In other research, we showed that amorphous powder can be an efficient photocatalyst, too [24].

There may be, however, even a better line of research for the cold spraying of photocatalytic coatings than the spraying of unmodified $TiO_2$. Previous efforts concerned mainly the physical parameters of feedstock powders, such as the degree of agglomeration or particle size [16,22,25–27]. What was not prioritized yet, is to consider the internal atomic structure and surface chemistry of the deposited powders. The wide bandgap of unmodified $TiO_2$ (approximately 3.0 eV for rutile and 3.2 eV for anatase) means that to obtain the activation energy for photocatalytic reactions, ultraviolet light is needed [1,2]. However, UV only constitutes ~5% of the solar energy that reaches the surface of Earth. To avoid artificial lighting and employ solar light, the latest scientific advancements in the field of photocatalysis aims to replace UV-activated photocatalysts with visible-light-induced photocatalysts [1,3]. Several strategies have been developed over the years in material sciences to alter the unmodified $TiO_2$ (both available on the market or self-synthesized). The main concept is to shift the absorption of $TiO_2$ towards longer wavelengths by doping the crystal structure with metal and non-metal ions [28,29]. Other ideas focus on dye sensitization [30] or the application of noble metals or other sensitizers as a cocatalyst to provide additional active sites [28,31]. The above procedures can be applied for already formed feedstock powder [29–31] or on the powder synthesis route [28]. All popular approaches rely on the interaction of new substances with $TiO_2$, which may lead to the formation of secondary impurities that could promote the adverse recombination of photogenerated charges ($e^-$–$h^+$). "Self-doped" $TiO_2$ with intrinsic point defects, such as vacancies and interstitials, might absorb energy from the visible region of the solar spectrum [32]. Such modified titanium dioxide is often no longer a white powder, however, it can turn black, grey or blue [32,33]. Although black $TiO_2$ is considered the most representative and effective for thermal hydrogenation (due to the absorption onset lying in the IR region and hence being activated with maximum solar energy) [34], its production requires complex synthetic methods. Moreover, oxygen vacancies and $Ti^{+3}$ ions generated in this high-temperature or high-pressure process can give the material high photocatalytic activity, yet can also play a negative role by becoming recombination centers (especially when their distribution is uncontrolled and not limited only to the semiconductor surface) [32,33]. Recently, it has been shown

that self-doping TiO$_2$ may be caused not only by oxygen deficiency manifested by various shades of black of the oxide but also by an excess of oxygen (induced by the treatment with oxidizing agents, such as hydrogen peroxide), which yields yellow-orange coloring of titanium dioxide [33]. Similarly as TiO$_2$ obtained using other absorption-shifting strategies, oxygen-rich yellow TiO$_2$ can be produced either through bottom-up synthesis [35–39] or by surface modification [40–42]. In both ways, sensitizing TiO$_2$ with H$_2$O$_2$ is simple as it lacks involving doping agents, metals, non-metals, etc. and hence has great application potential in the industry. Triangular peroxo–titanate complexes formed during synthesis with O-O bonds are predisposed to capture photogenerated electrons and use them for photocatalytic degradation, which reduces unfavorable recombination processes and significantly improves photocatalytic ones [35,42]. What is more, oxygen self-doping using hydrogen peroxide (H$_2$O$_2$) enhances the photocatalytic efficiency of TiO$_2$ in the form of crystal phases, such as rutile, anatase [37,40,42], and even amorphous phase [36,38,39]. To the best of our knowledge, only the oxygen-rich TiO$_2$ powders have been studied so far [35–42], however, there are no records of them having been used as a feedstock for coatings formation. Bearing in mind the high deposition efficiency of the amorphous phase in the low pressure cold spray process [16,23,24] and observing the excellent well-documented photoactive performance under visible light irradiation of amorphous yellow TiO$_2$ powders [35–39], we decided to employ oxygen-rich feedstock powder for cold spraying to investigate the potential of the spraying of visible-light active photocatalytic coatings.

Here we show a facile approach to obtaining the visible-light active titanium dioxide coatings from self-synthesized oxygen-rich titanium dioxide using low pressure cold spraying. First, we synthesized nanoparticulate amorphous oxygen-rich TiO$_2$ feedstock powder by one-pot hydrolysis of titanium isopropoxide (TiO$_2$ precursor) and its reaction with hydrogen peroxide (H$_2$O$_2$). Then, we sprayed as-prepared feedstock aiming at maintaining its high photocatalytic efficiency. Knowing that the activity of oxygen-rich TiO$_2$ is determined by the complex chemical nature of both the interior and the surface of the semiconductor, we conducted a Raman spectroscopy inspection to show how deposition parameters affect the defect chemistry of TiO$_2$. In the last part, we demonstrate the photocatalytic activity using the popular methylene blue dye as a model pollutant to ensure embedding activity in visible light.

## 2. Materials and Methods

### 2.1. Synthesis of Feedstock Powder

The amorphous oxygen-rich TiO$_2$ powder, referred to later as FP (feedstock powder), was produced in the one-pot synthesis. The reagents used were: 1 mL of nitric acid (HNO$_3$, 65%, Chempur, Karlsruhe, Germany), 100 mL of demineralized water, 10 mL of titanium (IV) isopropoxide (Ti[OCH(CH$_3$)$_2$]$_4$, 97%, Sigma-Aldrich, St. Louis, MO, USA) and 20 mL hydrogen peroxide (H$_2$O$_2$, 30%, Stanlab, Ltd., Lubin, Poland). Titanium precursor was added to the water-acid solution, and then the H$_2$O$_2$ was added to dope the already formed titanium dioxide with oxygen. A similar procedure was described elsewhere by Wu et al. in [35], with the difference being that in this research the titanium dioxide was air-dried at ambient temperature, not at 50 °C as Wu did. No color fading was observed after a year of storage in the air in room conditions.

### 2.2. Spraying of 200, 600 Coatings

As-synthesized feedstock powder was low pressure cold sprayed using DYMET 413 unit (Obninsk Center for Powder Spraying, Obninsk, Russia). Air with a pressure of 0.5 MPa served as a working gas and was accelerated in a circular de Laval nozzle with a throat and outlet diameters of 2.5 and 5 mm, respectively. Two various gas temperatures were tested: 200 and 600 °C. The samples produced in such conditions are referred to later as samples 200 and 600, respectively. A spraying gun was attached to the manipulator (BZT Maschinenbau GmbH, Leopoldshöhe, Germany) moving with a traverse speed of 2.5 mm/s and a stand-off distance of 10 mm. To ensure small waviness, the distance between the next



spraying beads was 2 mm. An untypical aerosol powder feeder (Palas GmbH, Karlsruhe, Germany) with a cylindrical chamber of 14 mm diameter, a height of 95 mm and a powder feeding rate of 61 g/h was applied. The powder was fed radially at the beginning of the divergent part of the nozzle by nitrogen with a pressure of 0.1 MPa. Aluminum alloy AW-1050A H14/H24 (min. 99.5 wt.% of Al) plates with dimensions of 20 × 20 × 4 mm were selected as a substrate material. Before spraying, the substrate surface was degreased, and grit blasted with alumina powder ($Al_2O_3$, mesh 45).

*2.3. Feedstock Powder and Coatings Characterization*

The crystal structure of feedstock powder and low pressure cold sprayed coatings was investigated using the X-ray diffractometer Ultima IV (Rigaku, Tokyo, Japan), with CuK$\alpha$ irradiation ($\lambda$ = 1.54056 Å) for the 2θ ranging from 5° to 75° in the 0.05 steps 3 s per each measurement point.

The evaluation of the surface and sections of the samples were conducted using the SEM microscope (Hitachi S-3400 N, Tokyo, Japan). For the cross-sections of coatings, the samples were cut in the middle of their lengths, embedded in the resin and consecutively polished without etching. The topography of the feedstock powder and coatings after spraying was investigated without any additional preparation.

The particle size analysis of the feedstock powder was carried out by laser diffraction using PSA-1190 (Anton Paar GmbH, Graz, Austria). After the initial measurement, the powder was subjected to ultrasound treatment for 30 min to observe the changes in the size of the initial agglomerates.

The surface roughness (Ra, Rz) of the coatings was measured using a profilometer (Form Talysurf 120 L, Taylor-Hobson, Leicester, United Kingdom). The diamond stylus with a radius of 5 μm was transversed at the contact mode with a measuring force of 0.75 mN and a measuring speed of 1 mm/s along tracing length $L_t$ = 15 mm and with a cut-off filter $\lambda_C$ = 2.5 mm.

The specific surface of the samples was measured using VHX-6000 digital microscope (Keyence, Osaka, Japan). The magnification was set to 200, which provided the surface area of 1.733 mm × 1.299 mm (2.233 mm$^2$). Three measurements were carried out for all deposited coatings and the mean value was determined for 200 and 600 samples.

The mass of the coatings was established by weighting the initial substrate before and after coating deposition.

Diffuse Reflectance Spectroscopy (DRS) was applied to determine the bandgap value ($E_g$) of the $TiO_2$ feedstock powder and coatings using a UV-VIS spectrophotometer equipped with a 75 mm integrating sphere (Specord 210, Analytik Jena, Jena, Germany). The DRS spectra of the sample were measured in the range of 200–800 nm with a Spectralon® as the reference material. The Kubelka–Munk function (Equation (1)) was used to convert the obtained reflectance ($R$) into the absorption coefficient ($F(R)$), and the Tauc's plot ($[F(R)h\upsilon]^{0.5}$ vs. h$\upsilon$) was drawn to determine the bandgap energy ($E_g$):

$$F(R) = \frac{(1-R)^2}{2R} \tag{1}$$

The Raman spectra of samples were collected using the Raman spectrophotometer LabRam HR800 (Horiba/Jobin-Yvon, Kyoto, Japan) upon Ar+ laser excitation at 514.55 nm with 50 mW laser power within the spectral range from 50 to 4000 cm$^{-1}$.

The photocatalytic degradation of methylene blue (MB) was carried out in the TOPT-V reactor equipped with eight quartz vessels with magnetic stirrers, and a low-temperature cooling circulating pump (Toption Instruments Co., Xi'an, China). The $TiO_2$ covered plates, immersed in MB solution (100 mL, $1\cdot10^{-5}$ M $\approx$ 3.2 mg/L), were exposed to VIS irradiation emitted from a xenon lamp (300 W) with a UV cut-off filter. The measured light intensity on the surface of $TiO_2$ was 0.31 mW/cm$^2$. To allow the equilibrium adsorption of MB on $TiO_2$, the process was initially conducted in the dark for 1 h. During the experiments, aliquots of the solution were collected every hour and the concentration of methylene blue was

measured using a UV-VIS spectrometer (Specord 210 Plus, Analytik Jena, Jena, Germany) at the wavelength of 668 nm (the detection limit was 0.25 mg/L with RSD $\leq$ 6%). Due to an insignificant pH decrease, no chemicals were used to maintain pH at a constant level. All studies (MB photocatalytic degradation as well as adsorption in the dark) were conducted in the photoreactor equipped with a circulation cooling system to ensure constant temperatures (21.5 °C) of all solutions in individual experiments during the tests.

## 3. Results and Discussion

There are a number of factors that influence the efficiency of the $TiO_2$ heterogeneous photocatalysis process: i.e., the degree and type of long-range ordering of atoms in the material, morphological properties that mainly determine the active surface of the catalyst, as well as the defects of its structure and surface. The latter factor seems especially important for oxygen-rich $TiO_2$, which has been intentionally defected to achieve an additional activity in visible light. Therefore, here we conduct research on the structural characteristics of the manufactured materials (powders and coatings), taking into account the morphology, the degree of crystal order, and the analysis of chemical groups modifying titanium dioxide (e.g., O–O and hydroxyl groups). The high photocatalytic efficiency of yellow powders is well reported in the literature [35–40,42], and therefore the main objective of the research is to preserve it after spraying to produce coatings. Here we discuss the feedstock powder characteristic first and then compare the data with the records for coatings sprayed using carrier gas at two different temperatures: 200 and 600 °C.

### 3.1. Crystal Structure Analysed Using X-ray Diffraction (XRD)

At first, we studied the phase composition of oxygen-rich $TiO_2$ in the form of powder and coatings. In the diffraction pattern of the feedstock powder (Figure 1, black plot), the broad hump centered approximately on 2 theta 25° indicates the amorphous structure of the powder. The diffractograms for coatings show that the amorphous share can be preserved during spraying in both samples (Figure 1, green and red plots). Spraying of the powder with carrier gas at 600 °C (Figure 1, red plot) led to partial crystallization of the employed feedstock powder. An asymmetrical anatase reflection (with Miller indices 101) at 25.35° appears out of the amorphous hump. Furthermore, the background of this sample is raised in the regions where other anatase peaks are located (consecutively 37.80° (004), 48.05° (200), 53.90° (105), 55.05° (211), 62.70° (204), 68.75° (116) (ICSD-9852, marked as letter A). The remaining, medium-intensity, peaks in the 600 diffractograms are present at 38.45°, 44.70° and 65.50° and originate from the aluminum substrate (ICSD-18839, marked as Al).

For the 200 coating (Figure 1, green plot), the diffraction of aluminum is stronger than in the case of the 600 sample. For example, the peak at 38.45° is approximately 100 times more intense than the background level (inset in Figure 1), which is a result of penetration of the aluminum substrate by X-rays during the diffraction measurement caused by the low thickness of the 200 coating. Apart from the aluminum in the diffractogram of the 200 sample, an intense peak centered at about 25° can be found, which may seem identical to anatase (detected in the red plot), however, clear differences exist between them: aluminum oxide peak is sharper and more symmetrical than building up anatase and the additional peak in the green plot is shifted toward the right to appear at 2θ equal 25.55° (ICSD-9770, marked as $Al_2O_3$). An additional confirmation of the presence of aluminum oxide is that at higher angles all subsequently observed peaks match the peaks characteristic of the alumina phase. $Al_2O_3$ can occur in the sample as the residue of substrate grit-blasting or as the passivation product. Here, the residual $Al_2O_3$ is more probable, as XRD studies of the substrate after grit-blasting confirm its presence. Interface contamination with blasting medium is a very popular side effect of grit blasting and can be omitted by using crystalline titanium oxide instead of $Al_2O_3$ [43]. That said, there are reports on the deposition of $TiO_2$ coating on $Al_2O_3$ substrates which increased their photocatalytic activity [44]. If passivation occurs while depositing a partially amorphous coating, it may provide better

adhesion of the coating to the substrate, via overgrowing, through the residual porosity of the coating [24]. However, the detection of both Al and $Al_2O_3$ may be evidence of the poor deposition efficiency in 200 sample.

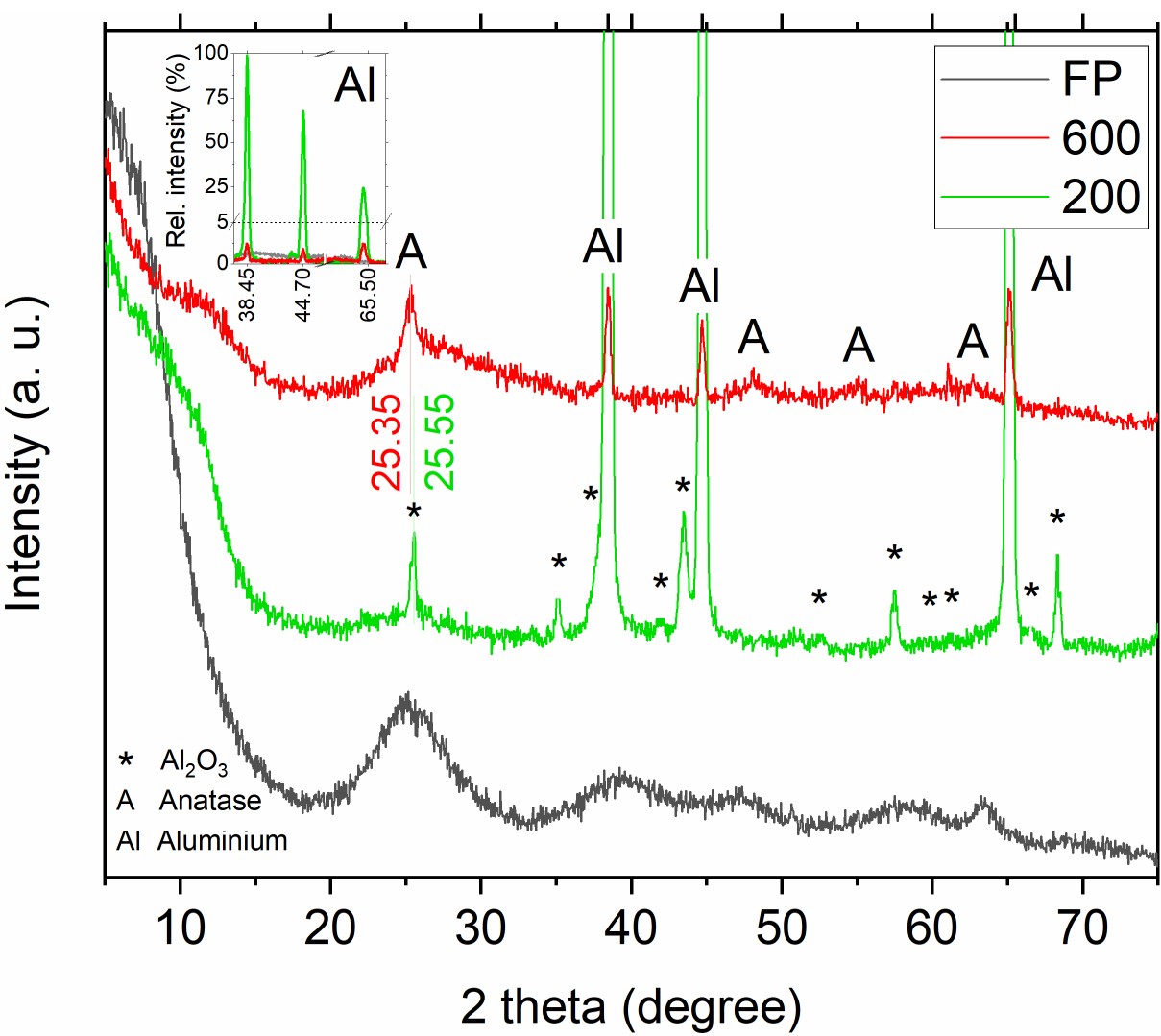

**Figure 1.** X-ray diffraction patterns of feedstock powder (black), sample 200 (green) and 600 (red).

The conducted X-ray diffraction tests showed that spraying the feedstock powder with cold gas has little effect on the long-range arrangement of the material. Even at a higher carrier gas temperature, the feedstock building coating crystallized only partially. At lower temperature, the deposited feedstock remained amorphous. We know from our previous work [23,24], that the amorphous form facilitated the deposition process and initiated the crystallization (observed here for the 600 sample), however, to determine the outcome for yellow $TiO_2$, further structural characterization is needed.

*3.2. Morphology and Microstructure Analysed Using Scanning Electron Microscopy (SEM) Supported via Particle Size Analysis and Roughness Measurements*

The results of XRD diffraction imposes, especially in 200 sample, a significant reflection that originated from the substrate material, encouraging the observation of the topography and cross-section of the samples. Again, we begin our observation by investigating the morphology of the powder. SEM micrographs (Figure 2) reveal strongly unsymmetrical agglomerates of $TiO_2$. The particle size of as-synthesized powder presented in (Figure 2a,b) is in the wide range of 6.8–212.2 μm (D0.5 = 44.1 μm). Regardless of the size of the

agglomerates (Figure 2c), they are porous and covered with smaller, densely packed unsymmetrical submicrometric flocculent particles. The agglomerates lacking flocculent covering seem denser, but still, they are characterized instead by the developed surface (Figure 2d). A 30-min ultrasound treatment for laser diffraction particle size analysis caused the detachment of the flocculent covering from the agglomerates, thus decreasing the size of the particles to 1.8–36.5 μm (D0.5 = 12.9 μm). This may suggest that the flocculent covering of agglomerates is weakly bonded, and hence it may be rather easily disintegrated, while in deposition (which proceeds in more drastic conditions than ultrasonication) (as in Figure 2d).

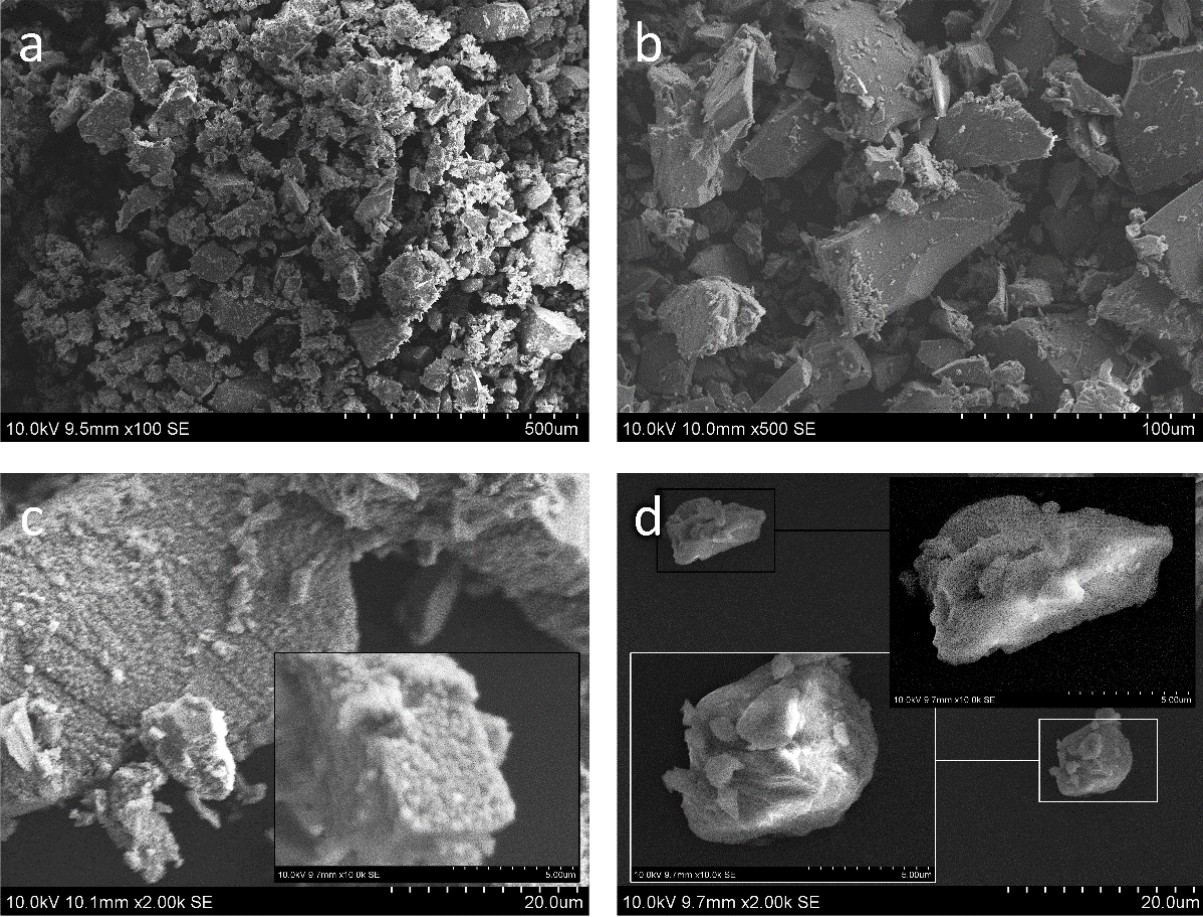

**Figure 2.** (**a**,**b**) SEM micrograph of feedstock powder morphology with a detailed view of (**c**) average as-synthesized agglomerates before DLS and (**d**) agglomerates after 30-min ultrasound treatment in DLS.

It is not only the particle interactions of a substrate that affects the deposition efficiency in cold spray; the surface roughness also has an impact. The irregular geometry introduced via grit-blasting (Ra = 8.84 μm, Rz = 51.72 μm) was developed to fit the size of the powder considered optimal for cold spray deposition. The initial parameters were modified upon the coating deposition process. The morphology of the 200 coating (Figure 3a) shows a rough surface with blunt edges of protruding irregularities. Since the roughness profile becomes more uniform after coating deposition (Ra = 5.91 μm, Rz = 37.12 μm) it would support the tendency of particles to fill the valleys of roughness and flatten only the top of the roughness peaks. However, at the magnification of 100 times, the coating is too thin to be successfully investigated (Figure 3c). On the contrary, the SEM morphology of the 600 sample (Figure 3b), shows an undulating surface. The substantial waviness is characteristic of cold sprayed coatings and results from the plastic deformation of the

surface upon anchoring large self-inflicted agglomerates [45]. The cross-sectional images (Figure 3d) reveal thick 25–50 µm dense coating, coarser with respect to the grit-blasted substrate (Ra = 12.18 µm, Rz = 67.49 µm). The roughness of the coatings favors the specific surface area. The measured specific surface of the 200 and 600 samples was 2.315 mm$^2$ and 2.456 mm$^2$, respectively. Compared to the surface area measured for the flat sample before grit-blasting (2.253 mm$^2$), the 200 showed an increase in the surface of 3% and the 600 sample—by 9%. The surface area of grit-blasted substrates was 2.366 mm$^2$, which means that the deposition of 200 coating flattened the surface, and the 600 coating—roughened it. Yet, at this magnification, good interlocking is observed throughout the entire cross-section.

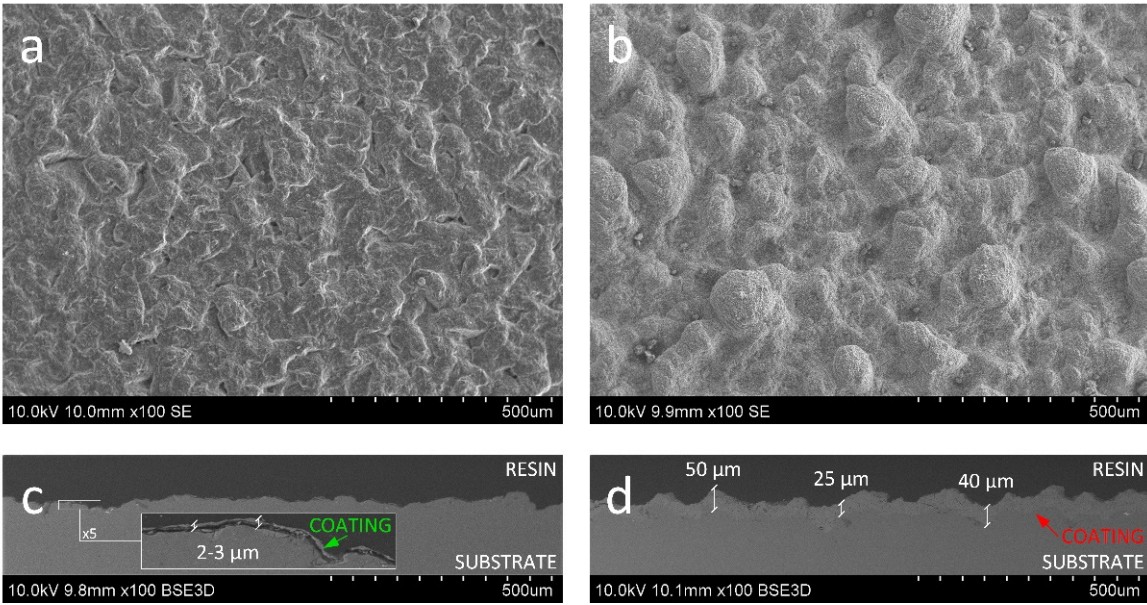

**Figure 3.** SEM micrograph of (**a,b**) topography and (**c,d**) cross section of (**a,c**) 200 and (**b,d**) 600 coatings. General view.

　　The morphology of the outer layer of both coatings (Figure 4a,b) has a feedstock powder-like structure. The top view of sample 200 resembles a shot-peened structure (Figure 4a). In the case of ceramic particles, the kinetic energy is transformed into the fragmentation of agglomerated particles instead of plastic deformation [45]. With this regard, probably only larger and denser agglomerates without flocculent cover reached the substrate and break apart upon impact. Generally, smaller particles, such as pieces of weekly bonded covering, accelerate more rapidly in the de Laval nozzle and achieve higher velocities, however, their kinetic energy can be lost in the bow shock region due to its lower mass [46]. The deceleration of the powder below the critical velocity contributes to the bouncing back of particles from the substrate and shock bow [26]. Hence, smaller flocculent particles are not observed in the 200 coating, which may be caused either by its bouncing back or its compaction by other incoming particles. The surface of the 600 sample seems more developed since the powder submicron structure was preserved (Figure 4b). The increase in the working gas temperature to 600 °C resulted in a greater acceleration of feedstock powder particles (even those flocculent covering—loose or weekly connected to the agglomerates) and therefore improved the impact velocity of particles and deposition efficiency, which is all consistent with the literature [27].

　　In both cases (Figure 4a,b), isolated discontinuities can be found (yellow arrows), much less frequently in the 600 sample, and hence the cross-sections were prepared to evaluate their severity (Figure 4c,d). In the case of the 200 sample, it was possible to form a very thin ceramic coating of 2–3 µm thickness (Figure 4c). The bond strength of the coating was not high enough to prevent detachment of the deposited particle by elastic spring-back forces during incoming particles bombardment [25]. The incoming particles significantly

densified the deposited thin ceramic coating, smoothing its surface upon the impact of the new particles. The thickness of the embedded coating was too small to absorb the excess energy, and, as a result, a net of cracks appeared in the coating material, leading to delamination of the coating. The cross-sections of the 600 sample (Figure 4d) show that it was possible to produce a uniform dense coating of high internal porosity and a rough surface well connected to the substrate. The outer part of the coating consists of loosely connected particles that were probably ejected from the gas stream and stuck on the coating surface. The detachments appear only in the upper part of the coating (yellow arrow, Figure 4d). The coating-substrate interface remains, however, solid.

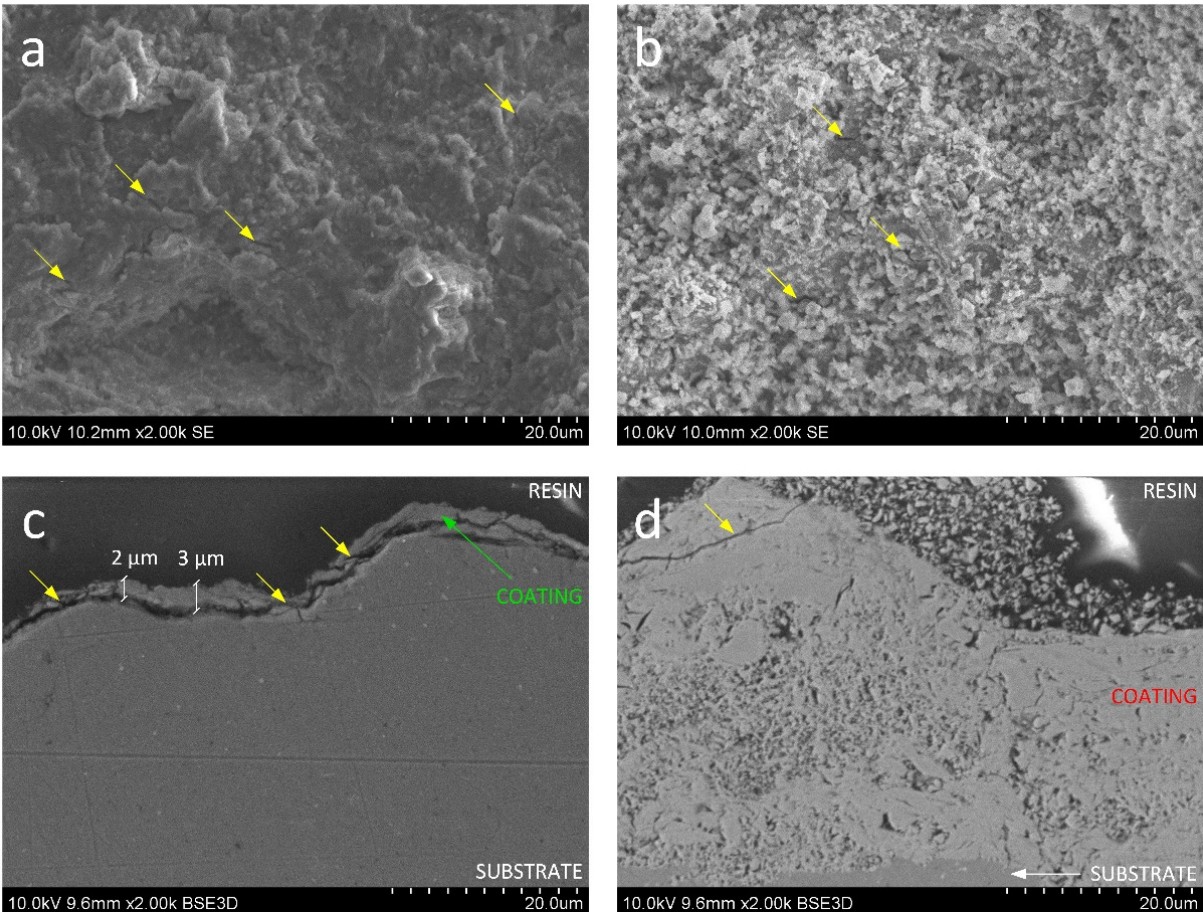

**Figure 4.** SEM micrograph of (**a**,**b**) topography and (**c**,**d**) cross section of (**a**,**c**) 200 and (**b**,**d**) 600 coatings. Detailed view.

The SEM analysis showed that in lower temperatures of carrier gas, only a very thin coating was possible to be deposited. Too low a level of both thermal and kinetic energy caused insufficient bonding of the coating to the substrate. Despite low adhesion, the coating was not fully detached and maintained contact with the substrate. The 600 coating was relatively thick and porous well connected to the substrate. The open porosity was the result of the agglomerates being broken apart and, as such, they revealed the internal porosity of the feedstock. Such reorganization enabled the surface of agglomerates to be compacted upon the impact on the substrate and the interior to be revealed. On one hand, the porosity may have been the origin of the crack formation, but in the case of the photocatalytic coatings, which were not under mechanical loading, the porosity ensures a high surface area.

### 3.3. Optical Properties Analysed Using Diffuse Reflectance Spectroscopy (DRS)

The optical properties of the feedstock powder and the cold-sprayed coatings were studied by UV-VIS diffuse reflectance spectra (DRS) (Figure 5). For all samples in the ultraviolet range, almost all incident UV light is absorbed (Figure 5a). The yellow color of the feedstock powder results in higher absorption at 400–500 nm visible as a shoulder to the peak located at ~320 nm (UV) [47]. The spectrally uniform absorption through the 400–800 nm range causes the yellow to fade, imparting simultaneously a grey-white finish [47]. The spectrum of the 600 sample shows a spectral response towards the visible region from 500 to 700 nm. The Tauc plots were used to estimate the bandgaps (Figure 5b). The Tauc plot for the feedstock powder reveals the presence of two optical bandgaps: one at 2.54 eV and another one at 2.24 eV. The records indicate that sample 200 has a lower bandgap (2.83 eV) than sample 600 (3.09 eV).

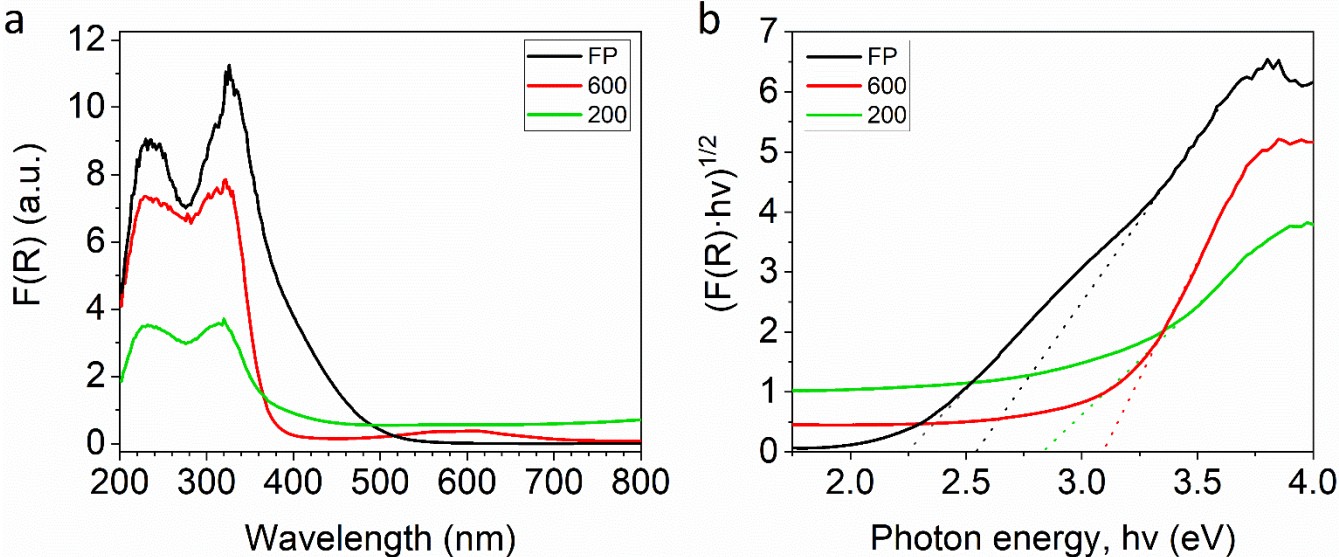

**Figure 5.** (**a**) Kubelka–Munk function spectra with (**b**) Tauc plot for estimating the bandgap energy for feedstock powder (black), sample 200 (green) and 600 (red).

The edge of adsorption is generally dependent on the crystal structure and the number of active sites (such as defects or dopants) [35]. In the literature, it is shown that the slight decrease in the bandgap could be, for instance, the result of oxygen vacancies and titanium ions present in the sample [48]. In other research, it is stated that the more defected state would lower the bandgap even more [49]. Here, the broadening of the bandgap of coatings with regard to feedstock powder must be directly related to the structural reorganization induced by spraying. The diffraction measurements (Figure 1) showed only a slight tendency to increase the degree of ordering of the titanium dioxide powder as a result of the spraying, while the optical measurements suggest that, in addition, there are also more subtle changes in the chemical structure. Therefore, it is necessary to investigate the structure and chemical composition of materials at the molecular level.

### 3.4. Vibrational Characterization Using Raman Spectroscopy

To gain additional information about the existence of the –O–O– coordination bonds in feedstock powder and coatings, the Raman scattering of all samples were measured (Figure 6). The summary of Raman analysis is displayed for convenience in Table 1.

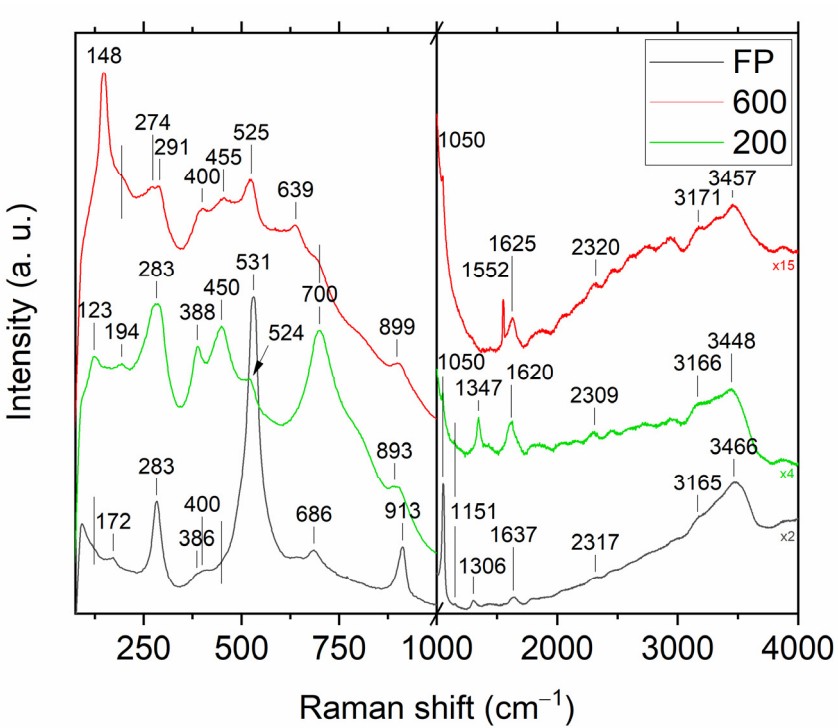

**Figure 6.** Raman spectra of feedstock powder (black), sample 200 (green) and 600 (red).

Due to omitting the drying step at elevated temperatures, we produced feedstock powder in the amorphous form (Figure 6, black plot). The amorphous structure of the powder has already been proven using XRD measurements (Figure 1), however the Raman spectroscopy is sensitive even to substantial changes in short-range interactions, which allows one to identify individual chemical bonds even in disordered amorphous materials. The low-intensity wide band found at about 400 cm$^{-1}$ was the only common band for the obtained spectra and anatase phase (B$_{1g}$), yet due to the fact that no stronger anatase vibrational mode (especially E$_g$ mode at 147 cm$^{-1}$ [50]) was observed in the spectrum, the formation of anatase in the feedstock powder was excluded. Furthermore, no characteristic bands for anatase could be found in sample 200 (Figure 6, green plot). The anatase structure (148, 194, 400, 525, 638 cm$^{-1}$ [50,51]), as identified by XRD (Figure 1), was detected only in a sample sprayed with carrier gas at 600 °C (Figure 6, red plot). If there is no evidence to consider crystal forms of TiO$_2$ in feedstock powder and given that in the Raman spectra of samples 200 and 600 not every band is explained so far, one may think that other modes originate from the interaction of Ti atoms with different forms of oxygen formed during synthesis [35].

The most intense band of feedstock powder spectra is located at 531 cm$^{-1}$. The mode observed in the range of 524–529 cm$^{-1}$ can be connected with the stretching vibration of Ti–O$_2^{2-}$, in which O$_2^{2-}$ is bound to a single Ti$^{4+}$ center in a side-on bonding configuration [52], which is called a triangular peroxy titanyl group in some works [53]. The presence of a band at this frequency is considered to be evidence of obtaining an oxygen-rich titanium dioxide powder containing O$_2^{2-}$ species incorporated during synthesis. When analyzing the spectrum of sample 200, a decrease in the intensity of this band is noticeable, which may indicate a partial loss of these active oxygen groups in the coating. In the 600 sample, it is not possible to detect the presence of a Ti–O$_2^{2-}$ band, as even if it exists, it overlaps with a relatively intense A$_{1g}$ anatase band at a frequency of 525 cm$^{-1}$.

In search of further evidence of significant modifications of the structure of titanium dioxide, it is worth looking at the band that appears in the case of feedstock powder at 686 cm$^{-1}$. This band is ascribed to stretching vibrations of Ti–O–O groups [54] or involving two-fold oxygen [55] in the literature. However, it is still unclear whether the Ti–O–O

vibration describes the vibrations of the peroxide group (i.e., the oxygen bridge –O–O–, $O_2^{2-}$) or the superoxide group (i.e., –O–O the end-on configuration, $O_2^-$, where two oxygen are attached to only one titanium ion). These two types of dioxygen groups differ significantly in the length of O–O bonds, which in Raman spectroscopy should cause a shift of the observed bands. Since the length of O–O bond in superoxides ($O_2^-$) is smaller than in the peroxide ($O_2^{-2}$) [56] groups, which corresponds to higher bond strength, a higher frequency of vibrations of the superoxide ($O_2^-$) groups can be expected. Moreover, in addition to the well-described peroxides and superoxides, there are species that have an intermediate character [56]. Compared to the state observed in feedstock powder, the intensity of the Ti–O–O band visible in the 200 sample significantly increases and shifts toward higher frequencies (700 cm$^{-1}$), which may indicate an increase in the content of Ti–O–O groups in the coating and, at the same time, reduce the distance between the interconnected oxide atoms, and thus increase the content of superoxide groups ($O_2^-$) in the coatings at the expense of peroxides ($O_2^{-2}$). The band at about 700 cm$^{-1}$ is still visible in the coating sprayed at 600 °C, although it is not as intense as the bands of anatase.

The Raman spectra of the measured samples contain two more bands in the frequency ranges, which in the literature are ascribed to the vibration of the triangular peroxy titanyl group, in which O–O vibrations [39,52,57] or Ti–O vibrations [53,54] occur. The O–O stretching in the feedstock powder is observed at 913 cm$^{-1}$, and in coatings, it blueshifts and loses intensity more for coating sprayed at higher temperatures. The intensity of the Ti–O vibrations in triangular titanyl groups (at ~1050 cm$^{-1}$) [53] is also reduced due to spraying. Therefore, all three bands assigned in the powder spectrum to the triangular titanyl groups (531, 913 and 1054 cm$^{-1}$) after spraying show a reduced intensity, indicating the thermal instability of the triangular peroxide species, which is consistent with the literature reports [53]. The disintegration of unstable peroxide groups is accompanied by the appearance of superoxide groups in the coatings, which is evidenced by an increase in the intensity of the 700 cm$^{-1}$ band in the spectra of the 200 coating. The second characteristic superoxide band [58] can be found at 1145 cm$^{-1}$ of the 200 spectrum. The superoxides are present also in the spectrum of the feedstock powder (~1152 cm$^{-1}$), which shows that $H_2O_2$ treatment forms not only peroxide groups. The typical band for superoxide groups (1145, ~1152 cm$^{-1}$) is not observed in the 600 coating, however, it is notable that only in the case of the coating sprayed with gas at a temperature of 600 °C can a very characteristic, sharp and relatively intense band at 1552 cm$^{-1}$, attributed to molecular oxygen ($O_2$) vibrations, be identified [59]. This observation implies that the active and fairly labile superoxide species present in the powder decomposes with the evolution of oxygen during the spraying of the 600 sample.

Since a correlation was found between the presence and number of hydroxyl groups on the TiO$_2$ surface and the photocatalytic potential (by helping to form radicals) [41], it is worth examining the measured Raman spectra in this regard. In all samples, broad high-frequency bands can be distinguished and usually assigned to OH groups, however, their intensity is the highest in the case of feedstock powder (green and red plots are y-stretched ×4 and ×15 times, respectively). The irregularity of the peaks may indicate several bands that overlap in that energy shift. The Raman spectra of all samples are characterized by the bands of OH group stretching and bending at ~3160–3450 cm$^{-1}$ (O–H stretching and bending vibrations, [59]) and deformation vibrations at ~1620–1640 cm$^{-1}$ (chemisorbed and/or physisorbed H–O–H [60]), which is evidence of a large amount of water molecules, adsorbed on the surface of titanium dioxide both in feedstock powder and in coatings. Another band associated with hydroxyl groups can be found at 283 cm$^{-1}$. It is easy to see that the width and the asymmetry of this band both increase after the spraying. In the literature on oxygen-rich TiO$_2$, vibrations in this frequency range are attributed to either Ti–O–H [40,41] or the intrinsic host lattice defects—oxygen vacancies [39,52], which are typical for oxides. Since spraying causes the appearance of two bands in the spectrum of the 600 sample, it can be assumed that the coatings contain both oxygen defects and

Ti–OH groups. Additional evidence of oxygen vacancies (in ordered structure) or oxygen deficiency (in disordered structure) can be found at ~450 cm$^{-1}$ [55,61].

**Table 1.** Assignments of the Raman bands in feedstock powder (black), sample 200 (green), and 600 (red).

| Description | Scheme | Observed Shift (Figure 6) (cm$^{-1}$) | | | Reported Shift (Literature Data) (cm$^{-1}$) |
|---|---|---|---|---|---|
| | | FP | 200 | 600 | |
| Anatase $E_g$ mode (symmetric stretching vibration of O–Ti–O) | O — Ti — O | | | 148 | 147 [50] |
| O–O vibration involving three- and four-coordinate oxygen | O — O | 172 | 194 | 194 | 160–240 [62] |
| Anatase $E_g$ mode (symmetric stretching vibration of O–Ti–O) | O — Ti — O | | | | 198 [50] |
| Ti–OH vibration | Ti — O — H | 283 | 283 | 274 291 | 282 [41] 284 [40] |
| Oxygen vacancy ($v_O$) (consequence of lack of oxygen) | Ti — $v_O$ — Ti | | | | 283 [52] 286 [39] |
| Ti–O bending and stretching vibration involving two-fold oxygen (X—unspecified atom) | Ti — O — X | ~382 | 388 | | 380 [55] * |
| Anatase $B_{1g}$ mode (anti-symmetric bending vibration of O–Ti–O) | O — Ti — O | ~400 | | 400 | 398 [50] 400–425 [63] |
| Ti–O bending vibration involving three-fold oxygen (consequence of lack of oxygen) | Ti — O | | 450 | 455 | 440 [55] * 440 [61] |
| Anatase $A_{1g}$, $B_{1g}$ mode (symmetrcic and anti-symmetric bending vibration of O–Ti–O) | O — Ti — O | | | 525 | 507, 519 [50] |
| Ti–$O_2$ symmetric stretching vibration of triangular peroxy titanyl group | O —— O \ / Ti | 531 | 524 | | 524–529 [52] |
| Anatase $E_g$ mode (symmetric stretching vibration of O–Ti–O) | O — Ti — O | | | 639 | 626–640 [63] 640 [50] |
| Ti–O–O stretching vibration (representing peroxo, superoxo or intermediate groups) | Ti — O — O Ti — O — O — X | 686 | 700 | ~700 | 667 [54] |
| O–O stretching vibration of coordinated peroxide ($O_2^{2-}$) species in triangular peroxy titanyl group | O —— O \ / Ti | 913 | ~903 | ~900 | 860–900 [57] 910–914 [52] 916 [39] |
| Ti-$O_2$ band vibration in triangular peroxy titanyl group | O —— O \ / Ti | 1050 | 1050 | 1050 | 1050 [53] |
| $O_2^-$ superoxide vibration | O — O$^-$ | 1152 | 1145 | | 1124–1148 [58] |
| O–O vibration in $H_2O_2$ | H — O — O — H | | 1347 | | 1347 [64] 1385 [59] |
| O–O stretching vibration in $O_2$ (molecular oxygen) | O = O | | | 1552 | 1552 [59] |

**Table 1.** *Cont.*

| Description | Scheme | Observed Shift (Figure 6) (cm$^{-1}$) | | | Reported Shift (Literature Data) (cm$^{-1}$) |
| --- | --- | --- | --- | --- | --- |
| | | FP | 200 | 600 | |
| OH bending vibration or scissoring of the chemisorbed and/or physisorbed water | H — O — H | 1637 | 1620 | 1625 | 1630 [60] |
| Precursor residuals vibration | CH$_3$ — C (CH$_3$, O) | 2317 | 2309 | 2320 | 2335 [54] |
| OH stretching vibration of hydroxyl groups from adsorbed water | H — O — H | ~3165 | 3166 | 3171 | 3050–3150 [59] |
| OH stretching vibration of H-bounded hydroxyl groups | H — O | 3332 | 3340 | 3315 | 3150–3500 [59] |
| OH stretching and bending vibration of surface mixed hydroxyl groups | H — O | 3466 | 3448 | 3457 | 3400–3600 [59] |

\* The referential data can be found in supplementary materials of the cited literature.

Considering the potential use of oxygen-rich TiO$_2$ coatings in photocatalytic reactors, the Raman measurements showed that the cold spraying process changes the chemical structure of the deposited powder. There is a visible decrease in the number of triangular peroxy titanyl groups and an increase in superoxide species could be crucial regarding the photocatalytic activity of coatings. The measured optical bandgap energy of the feedstock powder increases after the coating is deposited at 200 °C and experiences an even greater increase when the spraying temperature was 600 °C. In the literature [48,49], the differences in the bandgap width of the oxygen-rich and pure TiO$_2$ without additions are explained as a result of the introduction of oxygen defects into the structure of titanium dioxide, creating additional energy levels above the valence band. The Raman measurements showed that oxygen defects in feedstock powder and coatings have various forms (e.g., of superoxide groups, peroxide groups) and their number and type change depending on the parameters of coating deposition. These changes undoubtedly affect the electronic structure of TiO$_2$. However, the increase in the TiO$_2$ bandgap itself in no way determines the changes in photocatalytic activity. That is due to the loss of the peroxy groups in the coatings being considered to be responsible for the activity of the photocatalyst in visible light, which is compensated by the formation of superoxide groups known as oxidizing agents and radical initiators. Unpaired electrons make superoxides highly reactive, allowing them to oxidize various organic pollutants [65]. Many studies have shown that the hydroxyl groups on the metal oxide are responsible for trapping photogenerated charge carriers, resulting in a reduced rate of the recombination of electron-hole pairs [41]. Thus, the optimistic note is that spraying does not appear to significantly alter the level of hydroxylation of the titanium dioxide surface.

*3.5. Visible-Light Photocatalytic Activity via Photobleaching of Methylene Blue*

The analysis of the Raman spectra revealed that the coatings had a significant number of active oxygen species and hydroxyl groups, which suggests that the coatings should exhibit some photocatalytic activity in visible light. Although the bandgap is not directly connected to photocatalytic performance, it may be useful to select the activation light [41]. The bandgap of 200 sample was 2.8 eV, making the coating a promising candidate. Even if the 600 sample was characterized by a higher bandgap, with its crystal structure (mixed amorphous-anatase phase) it is a good candidate as well. It is important to emphasize that the masses of 200 and 600 coatings were considerably different, which is a result of

the various coating thicknesses (Figures 3 and 4). The 600 coatings weighed $35 \pm 9$ mg and all 200 coatings—were less than 1 mg. We summarize the results in Table 2. With this information, we performed the methylene blue (MB) degradation experiments under VIS irradiation to investigate the photocatalytic performance of 200 and 600 samples.

**Table 2.** Summary of the characterization of the samples 200 (green), and 600 (red).

| Sample | Phase Composition | Thickness [μm] | Bandgap [eV] | Coating Mass ** [mg] | MB Adsorption in DARK [%] | MB Degradation in VIS [%] |
|---|---|---|---|---|---|---|
| 200 | Amorphous | 2–3 | 2.83 | <1 *** | negligible | 3.1 |
| 600 | Amorphous-anatase | 25–50 | 3.09 | 35 ± 9 | 3.95 | 17.2 |

** Calculated as the difference between the mass of the coated sample and sample before spraying. *** The differences were lower than 1 mg.

The UV-VIS spectra of methylene blue before and after visible light irradiation (Figure 7a) show that the peak at 667 nm decreased markedly in the presence of the 600 sample (and only slightly in the case of the 200 sample), indicating the degradation of the auxochrome group of methylene blue, and revealing photocatalytic properties of the studied coatings after being irradiated with VIS light in the range of 450–650 nm (Figure 7b). As expected, both coatings exhibit some photocatalytic properties under visible light (Figure 7c,d, full markers). The 200 sample (Figure 7c) with a lower bandgap (2.83 eV) is substantially less active (3.1% MB degradation). The 600 sample (Figure 7d), characterized by the higher bandgap (3.09 eV), shows evident photocatalytic activity under visible light (17.2% MB degradation). To distinguish the adsorptive properties of both coatings from their activity, we performed additional tests in the dark (Figure 7c,d, empty markers). In the absence of the photocatalysts, no degradation of MB under visible light irradiation was observed. The MB adsorption efficiency (in the dark) was negligible for sample 200 and minor for sample 600 (3.95%).

The difference in the MB degradation efficiency and adsorption can be assigned to a substantially lower mass of the 200 sample (<1 mg, thickness 2–3 μm) than the mass of sample 600 (~34 mg, thickness 25–50 μm) and a different spatial structure. As the interfacial surface is primarily responsible for the effectiveness of all surface processes, the limited deposition of TiO$_2$ in the 200 sample resulted in a lower specific surface area, and finally in much lower photocatalytic activity. Due to the high porosity of both samples, the higher thickness of the 600 sample was more beneficial in terms of providing a higher specific surface area. Moreover, referring to the efficiency of the studied photocatalytic process, one can note that it was conducted under very low irradiance (0.31 mW/cm$^2$; $\lambda > 460$ nm), and for a small catalyst area (400 mm$^2$) immersed in 100 mL of the MB solution. Generally, with an increase in the intensity of radiation, an increase in the number of photons reaching the surface of the photocatalyst can be observed, which in turn causes the increase in the number of decomposed MB molecules. The enhancement of the photocatalytic performance of oxygen-rich TiO$_2$ can be the surface composition and size distribution rather than the lower bandgaps [41]. Nevertheless, in the presented results, both coatings, which remained mechanically stable after the test, demonstrated the photodegradation of methylene blue. In the case of the 200 sample, it is important to provide an adequate surface specific area, which is also possible using the cold spray method and together with a lower bandgap could impart higher photocatalytic activity. The obtained results reveal the high photocatalytic activity of the 600 sample obtained by the proposed cold spray method and make further research relevant and promising.

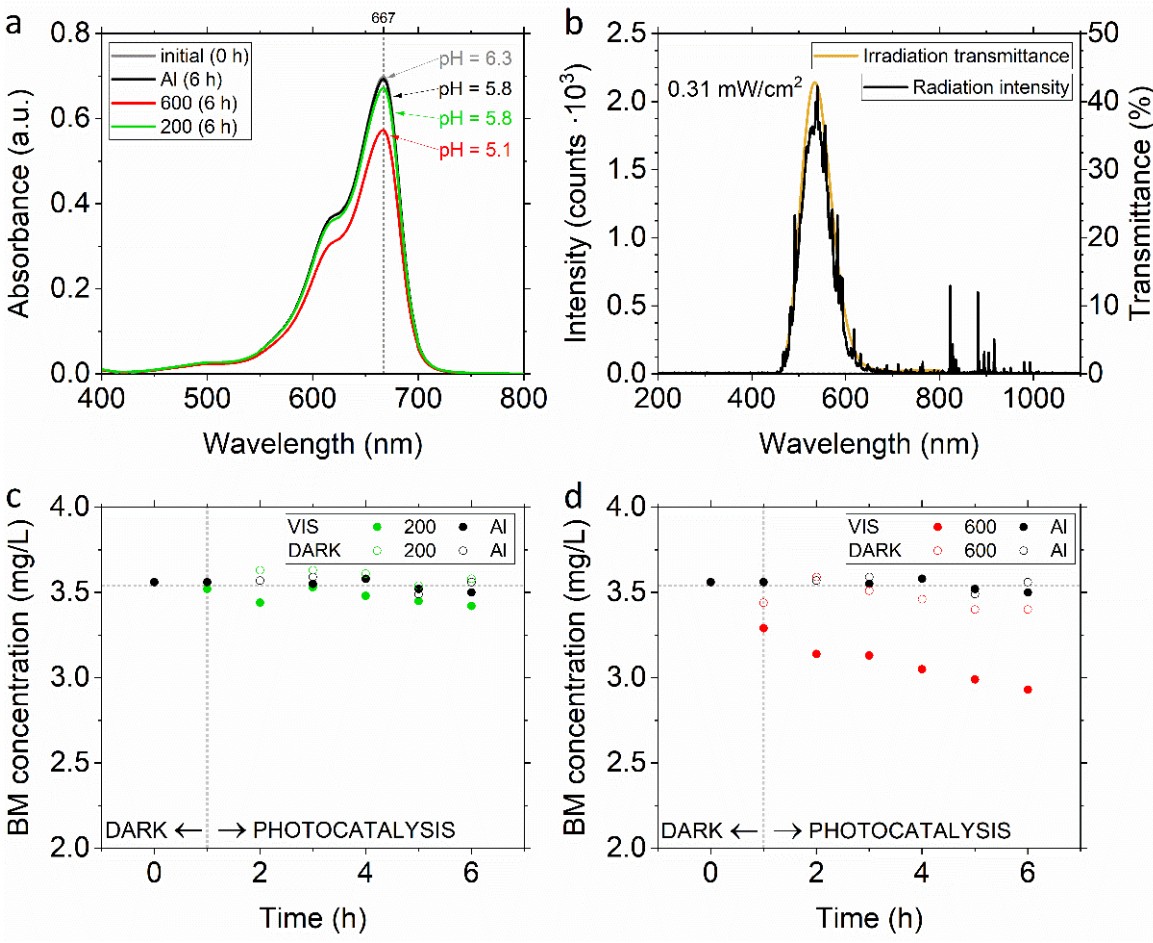

**Figure 7.** (**a**) Absorption spectra and (**c**,**d**) kinetics of methylene blue (MB) degradation in the presence of sample 200 (green) and 600 (red) and inactive Al substrate (black) under visible light irradiation from Xe lamp equipped with the UV cut-off filter (**b**) with its characteristics: measured radiation intensity (black), irradiation transmittance through the UV cut-off filter measured spectrophotometrically (yellow).

## 4. Conclusions

In this study, we low pressure cold sprayed amorphous oxygen-rich titanium dioxide to produce the coatings exhibiting photocatalytic activity in visible light. We analyzed changes in the feedstock powder before and after spraying to understand how the selected parameters influence the efficiency of the heterogeneous $TiO_2$ photocatalysis process. Using carrier gas at 600 °C, we deposited a thick (25–50 μm), porous coating with a highly developed surface. Spraying with gas preheated to 200 °C led to the formation of a relatively thinner (2–3 μm) coating with a net of discontinuities. We showed that both coatings were effective in the degradation of methylene blue upon the visible light irradiation, and the morphology of both has not changed after the photocatalytic efficiency test. Hence, the chosen powder immobilization method (LPCS) selected as a result of the efficiency of deposition of amorphous materials turned out to be safe for the subtle, yet substantial, physicochemical structure of the oxygen-rich feedstock powder. Consequently, low pressure cold spraying not only represents low-temperature large-scale technology; it also allows for the preservation of the feedstock powder absorption edge extended to visible light.

**Author Contributions:** Conceptualization, A.G. and A.B.; methodology, A.G., A.B. and D.O.; validation, A.B., M.J., M.W.; formal analysis, A.G., A.B., D.O., M.W. and M.J.; investigation, A.G., A.B., M.J., D.O. and M.W.; resources, M.J., M.W. and D.O.; data curation, A.G., A.B., D.O.; writing—original

draft preparation, A.G., A.B., M.J., M.W. and D.O.; writing—review and editing, A.G. and A.B.; visualization, A.G.; supervision, A.B.; project administration, A.G.; funding acquisition, A.B. and M.J. All authors have read and agreed to the published version of the manuscript.

**Funding:** This research was funded under statutory activity subsidy from the Polish Ministry of Education and Science for Wroclaw University of Science and Technology (K58W10D07, K60W10D07) under grant number 8211104160.

**Institutional Review Board Statement:** Not applicable.

**Informed Consent Statement:** Not applicable.

**Data Availability Statement:** Not applicable.

**Acknowledgments:** A.G. acknowledges special thanks to Irena Jacukowicz-Sobala, Department of Chemical Technology, Wroclaw University of Economics and Business, for her involvement and guidance in UV-VIS measurements.

**Conflicts of Interest:** The authors declare no conflict of interest.

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
