# Peer review of "Preparation of Visible-Light Active Oxygen-Rich TiO2 Coatings Using Low Pressure Cold Spraying"

_coatings, doi:10.3390/coatings12040475_

Round 1
Reviewer 1 Report
The following interesting paper entitled “Preparation of visible-light active oxygen-rich TiO2 coatings using low-pressure cold spraying” discusses photoactivity of modified amorphous TiO2 coating that synthesized through an eco-friendly, cost-efficient, and low energy consumption approach, known as low-pressure cold spraying. This study has novelty and the manuscript is well structured and reads well overall, although it will need a spelling check. I suggest this article be published after a minor revision.
The abstract is clear and concise.
The introduction is informative and rational.
The conclusions are logical.
- I strongly recommend authors sketch a simple graphical abstract for your article. Indeed, an informative well designed graphical abstract allows readers to gain a better understanding of the big picture at a glance. Moreover, it can help the visibility of your paper.
- In Figure 2 parts “a” and “b” in which authors discussed particle size distribution, I suggest using ImageJ software (line tool) to better show particle size distribution in each SEM image. Right now, it is not easy to match the numbers mentioned in the text with particle size in the image.
- The authors of this paper believe that the photoactivity of synthetic TiO2 is heavily dependent on the specific surface area than their energy bandgap while no data have been presented regarding the value of the specific surface area of these two samples (200 and 600). Please provide surface area data to strongly defend this hypothesis.
- In Figure 1, since you have the legend inside the figure that shows which color is related to which sample, you don’t need to repeat it all again in the figure caption (same comment for the rest of the figures). Moreover, you don’t need to repeat the full name of FP (feedstock powder) here or anywhere else in this paper since you have already mentioned it in the “Materials and Methods” section. Similarly, please edit it for the rest of the manuscript.
- In Figure 1(XRD spectra), please edit the unit of the y-axis to “Intensity (a.u.)”. Same comment for the Raman spectra (Figure 6).
- In Figure 2, the caption is kinda weird! Please revise it (something similar to other captions like Figures 3 and 4). Please keep it consistent with the other captions in the whole manuscript.
- Figure 5 is not legible and kind of messy! Please resketch the Tauc plot and rearrange graphs in a way it can be clearly seen by the reader of your paper.
- In Table 1, the Raman shifts are unitless! Please write down (cm-1) in the first row like
Observed Shift (cm-1), Reported Shift (Literature Data) (cm-1)
- Since parameters such as temperature and solution pH can affect the adsorption and photodegradation process, please specify in which condition have performed your experiment.
- Please be consistent in using units in the whole manuscript. For example, in the case of concentration, you have used “Molarity” in line #178 and “mg/L” in Figure 7 as well.
Author Response
Dear Reviewer 1,
all the authors would like to thank for all the time and effort put into reviewing the paper. Given the professional critique, we decided to introduce several modifications aiming at the improvement of the substantive and aesthetic value of the presented paper. The particular concerns are examined separately and the responses are submitted in the file enclosed below. It is our belief that we managed to satisfy the requirements of the manuscript acceptable for publication in the special issue of the Coatings.
Yours sincerely,
Anna Gibas on behalf of the authors

Reviewer 2 Report
The manufacturing is focused on developing an oxygen-Rich TiO2 coating. The objective is well defined, and the manuscript is well written—the following need to be addressed.
- Many of the statements needs references, for example, line 41-43 and line 71-73.
- From Fig 3C, it is difficult to appreciate if the coating exists. It would be better if the author could provide a higher magnification image clearly showing the coating on the base.
- In Fig 4d, please indicate coating and the base using an arrow
- Authors indicate a lower bandgap of sample 200 than sample 600. Is there any effect of Al2O3 on the bandgap in the sample? Please explain.
- It is unclear what structural characteristics dominate in changing the bandgap. Please explain in detail.
- In conclusion, the authors suggest the coated samples are effective in the degradation of MB. How the efficiency of the coated samples in degradation of MB as compared to the conventional coated TiO2. A comparison value will give an idea of the proposed coating sample efficiency in the degradation of MB.
Author Response
Dear Reviewer 2,
all the authors would like to thank for all the time and effort put into reviewing the paper. Given the professional critique, we decided to introduce several modifications aiming at the improvement of the substantive and aesthetic value of the presented paper. The particular concerns are examined separately and the responses are submitted in the file enclosed below. It is our belief that we managed to satisfy the requirements of the manuscript acceptable for publication in the special issue of the Coatings.
Yours sincerely,
Anna Gibas on behalf of the authors

Reviewer 3 Report
Overall, the work appears to be solid and accurate. However, some major revisions shall be done by the authors to make it publishable.
- The first three lines in the abstract contain some English inaccuracies and, more importantly, can be omitted entirely as they do not suit the purpose of an abstract, i.e. summarize what has been done in the work. It is perfectly ok to start the abstract by just saying "In this paper we show a facile approach to low pressure cold spray...".
- The main scientific subject on which the manuscript focuses on appears to be the realization of visible light-activated photocatalyst films on solid substrates via spray coating. The possibility to achieve efficient visible-light activated photocatalytic transformations and oxidations is a crucial topic, as correctly discussed by the Authors. There are several synthetic approaches for modifying the titania so that it becomes active under visible illumination. These approaches, in most cases, modify the optical absorption spectra and give some "color" to the otherwise white TiO2 powders. Oxygen rich ("Yellow") TiO2 is one of the materials involved in this subject, even though not the only one and, probably, neither the most promising one. I suggest that the authors mention this point by mentioning and giving additional references on other "colored versions" of TiO2, which may include for example the "blue" and the "grey" (or black) TiO2. Some suggested references (just four) to include in the additional part of the revised introduction:
Blue Titania:
- Phys. Chem. C 2020, 124, 3564−3576 (https://dx.doi.org/10.1021/acs.jpcc.9b08993)
- -ACS Catal. 2012, 2, 2641−2647 (dx.doi.org/10.1021/cs300593d)
Black and/or grey Titania:
- Nanoscale 7, 45, 19184-19192 DOI: 10.1039/c5nr05975e
(This latter is of special relevance for this manuscript also because the authors discuss about the switching from oxygen rich yellow anatase to oxygen vacancy rich black anatase TiO2)
- Chem. Soc. Rev., 2015, 44, 1861-1885 DOI: 10.1039/c4cs00330f
This is just a minimal group of four high-quality works spanning he recent years. It they wish, the authors might add few others. A reader would like to know also about these other approaches.
Also, the authors might attempt to explain or argue which are the advantages (and disadvantages, eventually) of the approaches based on preparing O-rich titania instead of other forms of modified TiO2 (as, for example, the other "colored" version mentioned).
- It is quite important, for the sake of evidencing the actual performances of the samples, to clarify the spectrum of the light used to activate the O-rich TiO2 films. Hence, the authors shall either provide the actual intensity optical spectrum (if they have tools/instruments to measure it) or at least give a reference spectrum of the unfiltered light (which is usually provided by the lamp manufacturer) plus mentioning which optical filter was used to cut off the UV light. Even a small fraction of light in the range close to 400-420 nm can make a lot of difference. If the authors can provide an experimental spectrum, it shall be added in the manuscript (it is an important information).
- I see no reason in showing the plots of diffused reflectance spectra instead of the Kubelka-Munk functions (i.e. optical absorption) spectra. Hence, I recommend the authors to change (expand) the Figure 5 by adding also the Kubelka-Munk spectra (i.e. F vs optical wavelength). This would give a much better insight on the optical properties of the samples.
Moreover, I would suggest showing the bare Kubelka-Munk spectra without Tauc plot for two reasons: 1) the Tauc fit is already reported in Figure 5 and 2) many scientists rely more on seeing the actual absorption spectrum than on the Tauc fits and Tauc gaps.
- The disparity in the amount of catalyst used in the two cases (Figure 7) makes it almost useless to report the data for the 200°C treated sample and to compare it with the 600°C sample. I would recommend to show only the latter in Figure 7, as the comparison might give the impression that the first material (200°C) is inactive and useless, but we actually do not really know if it is true or not.
After the revisions, I think that the re-evaluation of the article shall be positive and that it can be expected that the revised manuscript gets published.
Author Response
Dear Reviewer 3,
all the authors would like to thank for all the time and effort put into reviewing the paper. Given the professional critique, we decided to introduce several modifications aiming at the improvement of the substantive and aesthetic value of the presented paper. The particular concerns are examined separately and the responses are submitted in the file enclosed below. It is our belief that we managed to satisfy the requirements of the manuscript acceptable for publication in the special issue of the Coatings.
Yours sincerely,
Anna Gibas on behalf of the authors

Round 2
Reviewer 3 Report
I think that the revisions are satisfactorily. Therefore, I recommend to accept the present version of the manuscript for publication.
Author Response
Tnak you.